# A Multi-Modal Approach for Depression Detection

## 1   Introduction

Depression is a widely common mental illness and one of the leading causes of disability worldwide, affecting 5% of the global adult population (on average). [1] With 280 million people being afflicted by the condition worldwide  [1] With 40% of Canadians with mental illnesses seeking professional help, exploring and refining effective screening methods to reach wide audiences can help reduce the amount of people who continue to suffer from mental illness without knowledge, community support, or the assistance of a mental health professional.  [4] In our study, we build a framework based on graph neural network to predict the mental status of a twitter user. Experimental result has shown that our model obtained a stable performance in the real life scenario.

## 2   Literature Review

De Chouhudry et. al  [3] examine the potential for leveraging social media as a tool for understanding the distribution of depression in populations, and using probabilistic models to determine if posts could indicate depression.  The authors the CES-D self-reporting results  [9] and twitter feeds of 489 results to perform their analysis. Using emotions inferred from post text, time, linguistic styles, n-grams, user engagement with others in the platform, number of people following a user, and the number of people a user follows as features, the authors were able to demonstrate the potential of using social media as a reliable tool for measuring depression distribution at a broad population level.

Tsugawa et. al. [11] also the effectiveness of using social media information and activity as a tool for predicting if a user suffers from depression, and their approach for prediction uses Support-Vector-Machine (SVM) as an algorithm to predict whether or not a Twitter user has depression.  Before beginning prediction, the authors released a website containing a questionnaire for volunteers to sign up and answer questions on demographics (age, gender, socioeconomic/health status) and use the CES-D self-reporting scale for depression [9] to guide their evaluation of whether or not a user has depression. The information obtained from the volunteers by this questionnaire would serve as the ground-truth in their approach. The authors conclude that topics estimated with topic models inferred from 2 months of twitter activity and word frequency served as useful features in their predictions, and that the text features obtained from user activity can be used to make predictions with 69-percent accuracy. Although they focus more on a balanced setting where the same number of depressed and non-depressed users is considered, the authors also discuss other distributions of the two classes, ranging from 10-90%, with the balanced classification giving the highest accuracy score.

Gui et. al [5] expand on this by including images as features in their dataset, and use reinforcement learning instead of an SVM. The authors propose the use of multi-agent reinforcement learning with multiple policy gradients to select texts and images to evaluate the usefulness of the joint-action of these agents in predicting if a user has depression, based on the accuracy of their classifications. This gave them an accuracy of 90-percent (0.900). The authors were able to improve the accuracy of their predictions when compared to the accuracy of the text-based-only or image-based-only approaches cited in their work. The authors propose COMMA policy gradients which use a centralized training framework with decentralized execution. This is done by using a centralized critic for image and text selection, which have differentiated advantages [5]. Text features were extracted by Gated Recurrence Units (GRU) [2] , and image selection was done by a pre-trained Visual Geometry Group (VGG) network  [10]. Despite expanding on the number of features available, the authors also handled a balanced setting.

Orabi et. al [6] presented a novel approach to optimize word-embedding for classification, using the CLPsych 2015 Shared Data Set.  [8], a data set of 1,145 twitter users labelled by control group, depressed group, and PTSD group, as well as the Bell Let's Talk data set The information includes number of followers/people following, twitter users and

35th Conference on Neural Information Processing Systems (NeurIPS 2021), Sydney, Australia.

# 3 Methodology

## 3.1 Data Collection

Use Twitter's API to scrape Twitter users publicly available tweeting history and profile information. Each individual Twitter user will be a labelled data-point inside out data-set. The Twitter user will be labelled as "depressed" group if they have a tweet that contains a string that matches or is similar to "I have depression/ I am diagnosed depressed". If the Twitter user's profile does not contain this string inside any of their tweets their profile will be labelled as "non depressed".

Stored in each data-point representing a individual Twitter user will also be the entirety of their textual tweeting history, their media tweeting history, their profile photo and their Twitter bio.

Our data collection approach will allow us to identify which Twitter user has connected with what other Twitter users by scanning their textual tweet history for interactions with other users. The data of interacted user will also be collected to complement our user network. We will use this information to construct a graph that models the interactions between all of the Twitter users inside of our data set.

After finishing the first round of data collection, a serious problem was found that the positive case (i.e. depressed user) rate is only 0.004. Among a sample of 1200 twitter users, only 6 users were labeled as positive. To make the data more meaningful to be learned, we modified our labeling standard. Instead of using keywords detection, we utilize sentimental analysis to conduct classification of users. Any user with more than 10% extreme negative tweets (having compound score less than -0.6) is considered as a positive user. Under this method, the positive rate reached to 20%.

## 3.2 Baseline Model

In this report we proposed a new multi-agent model which takes user's tweet text, image and social media network to make prediction. To compare performance with other similar models, we select the Cooperative Multimodal in Gui's work [5] as baseline. Gui's model is an Actor-Critic reinforcement learning with one Text Actor and one Image Actor. To obtain image and text features used two feed these two Actors, the baseline also contains a 16-layer pre-trained VGGNet model to extract image feature and a GRU to compute continuous representations of sentences. At each step, the two agents compare the current global reward to the reward received when that agent's action is replaced with an opposite action.

## 3.3 Algorithm Development

We expand on Gui's work [5] where a multi-modal reinforcement learning approach was used to predict whether or not a user was depressed based only on their textual tweet and photo tweet history.

## 3.4 GNN

Graph Neural Network (GNN) is a type of Neural Network that directly runs on graphs. In the social network scenario, users are represented as nodes and relation between users are represented as edges. Therefore, our task is to classify each node as depressed or not depressed. There are three main components in the graph, they are defined as follows

- edge_index: This is a matrix representation of the connections between users and is stored as Coordinate Format(COO). The matrix has size of ($2\times$ number of edges). E.g., if we have $[[1, 0, 2], [3, 1, 1]]$, then there are three edges, (user1,user3), (user0, user1), (user2,user1).

- The input $X$: There are no standard definition of $X$. Since our task is node classification, we have defined our $X$ to be embedding of users which means it has the size of (number of users $\times$ features). For further discussion, see section 3.5.

- The label $Y$: This is a Boolean-valued vector of the each node, represents if the user(node) is depressed or not.

Then, the GNN we're using is Graph Convolutional Network(GCNConv)[7] and some message passing technique is used.

### 3.5 Node Embedding

As described as the previous part, we use a giant graph to represent the relationship between users on social media platform, we need a compact representation of each user in the model. In our work, each user is represented by a 384-dimension vector. to construct the user embedding we did the following. For each user we extract all her past tweets and perform some prepossessing: (i)Stopwords removal, lemmatization and etc; (ii)Then we filter out only the tweets with length greater then 5 words and concatenate all of them into a single string; (iii)At last we use the sentence-transformer to compute the word2vec embedding as a 384-dimension vector. The above embedding process allows our model to have a high discriminating capability.

## 4 Experimental Evaluation

In this section we will report the process that we conducted our experimental design and how we evaluate our model.

### 4.1 Experimental Setting

We evaluate our model on two datasets that was described in the data collection section. The first dataset has 27000 users and the second one has 2260 users. we also evaluate on he test-dev split where we split the first dataset in two two subsets and 70 % used for training and 30 % for testing. For evaluation metric, we will primaryly focus on accuracy in our remaining sections.

### 4.2 Training

We trained our GNN model on a RTX GPU. Our model were trained for 500 epochs using batch gradient decent with batch size = 128 with weight decay. During the training process, the major issue we found is that the target classes are imlanced and our model is biased to the majority class. Data augmentation has also been applied in the training process, we used the GraphSMOTE[12] to solve the imbalance issue. The hyper-parameters(GNN layer type, inner-layer number, loss function) were manually picked on the dev set.

### 4.3 Results and Discussion

The best result we obtained is 87 % on training set and 85 % on test set. We empirically analyze the evaluation of the number of hidden layers to the model and found that the current model is the best among the search space.

## 5 Conclusion

In this study, we present a framework based on Graph Neural Network to predict the mental status of a twitter user. We demonstrate that the graph can efficiently represent the topology among the users. Through several experiments, we also found that our model obtained a strong performance in real life scenario.

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
