# OpenReview forum: "A Multi-Modal Approach for Depression Detection"
_uoft.ai/University_of_Toronto/2021/ProjectX — Submitted to ProjectX2021_

### Official Review · Reviewer_DS1Q · 2022-02-09
**Good starting point for a machine learning approach to using social media to predict depression - Results/discussion sections need much more work**

**Rating:** 5
**Confidence:** 3

**Review:**

Connection to current science: 2/3
* Great work justifying the importance of depression and summarizing its impacts
* Nice summary of previous work looking at predictors of depression from social media
* It would have been ideal to include some information on what this paper adds to the existing literature. Does it address any knowledge gaps or limitations from previous papers?
* More information is needed on potential applications and implications. What are the authors’ thoughts on how predictive models for depression using social media could eventually be implemented? What would be the implications (e.g., population health impact, challenges, etc.)?
* The results/discussion section is only two sentences long and needs much more work. This would normally not be acceptable for a paper, but some leniency is given here because of the time constraints of this project.

Clarity of communication: 1.5/2
* Up until the results/discussion section, the paper is clear, easy to follow, and enjoyable to read.
* No tables or figures were presented, but it would have been interesting to see the data and/or results visualized in some way.
* The paper is presented in a logical structure
* It is not clear what metric the 85% and 87% reported in the results refer to. This is very ambiguous and needs to be clarified.

Methodological quality: 2.5/4
* The authors did not describe how the sample of Twitter users was selected. This is a key omission, as it may help provide insight into potential selection bias (affecting the generalizability of the results).
* No description of limitations associated with including only Twitter users
* The authors justified changing their definition of a positive case to >10% extreme negative Tweets, but it is not clear how or why this threshold was chosen. It would have been interesting to see results from a sensitivity analysis examining how the results change with different thresholds.


Reproducibility: 0.5/1
* Good work considering it was conducted over a period of 5 months
* The authors did not describe how their sample was selected, and made no mention of their code being available for others to access. If any of this is available, it should be stated in the paper.

---

### Official Review · Reviewer_ZP5R · 2022-02-10
**Multi-modal approach for depression detection review**

**Rating:** 6
**Confidence:** 4

**Review:**

Connection to current science (1.5/3):
- Novelty of this approach is unclear - what gap in the literature are you attempting to fill?
- Relies on other models to classify whether twitter users are depressed - how would this affect the results? Bias?

Methodological Quality (3/4):
- The method of identifying depressed twitter users could have been evaluated more carefully.
- Methods are appropriate, but were any other metrics considered beyond accuracy?
- Assumptions are reasonable.

Communication (1/2):
- A number of spelling and grammar errors.
- Reference numbering out of order
- No figures to verify the numbers reported as results....how can you claim that the number of layers you selected is the best without presenting the evidence of this?

Reproducibility (0.5/1): This work would be quite difficult to reproduce based on the information provided.

---

### Official Review · Reviewer_tbg5 · 2022-02-14
**Response to multi-modal approach for depression detection**

**Rating:** 4
**Confidence:** 3

**Review:**

Connection to Current Science:
The authors have not convinced me through their writing that implementation would make a significant contribution to the field. This could be due to important key words missing in their introductory paragraph, or omitted important details in the literature review. E.g., who are the users of twitter, how does this apply outside of twitter data?
The objective of the present study is not explicit. There is a single sentence in the introduction which provides an insight, but it is on the reader to interpret this. A paragraph following the literature review which outlines the overall aim/objective of the present study would aid the reader.
Results and discussion should be two separate sections.
What do the results mean (e.g., 87% and 85% of what? Tell your reader in the results).
In the discussion, indicate how these results are relevant – what does one do with this information?
It’s not clear how these results are impactful.

Clarity of Communication:
Introduction typically introduces the topic and positions it as important in the literature. This
Literature review/background should define important terms e.g., CES-D.
Numerous grammatical errors interrupt the flow e.g., incomplete sentences, omitted words, Typos/spelling errors, lack of capitalization.
References are incomplete.
Sections of the paper include material which should be include elsewhere e.g., introduction includes results/concluding statement, results of data collection included within the data collection (instead, describe how you QC’d the data).


Method Quality:
Describe sources of bias: use of twitter data, use of only public twitter data.
Positive case definition was modified - what does a positive user mean, if you’ve changed the definition? How is this valid?
Provide more details on the training process.
Given the limited details on results and discussion it is not clear how this work improves on previous methods or advances the field.
The authors make statements that are not clearly supported by the methods, as they have drafted e.g., “We demonstrate that the graph can efficiently represent the topology among users.”

---

### Official Review · Reviewer_6Lya · 2022-02-14
**Overall good project but with gaps**

**Rating:** 4
**Confidence:** 4

**Review:**

Connection with current science: 2 out of 3
The authors base their work with respect to a recent example of a well published article. Using this as example they review the literature and motivate their problem very well. They also discuss some of the challenges that can be expected.

Clarity of Communication 1 out of 2
The paper reads well but the presentation could have been made more amenable. Some figures in the main submission would have made a difference

Methodological Quality 2 out of 4

The authors struggled with a technical issued found after having set definitions that implied an outcome that was very rare to be well predicted. The authors suggested changed to tackle this which were innovative (and somehow acceptable but debatable). However, the discussion of results was poor and I would have liked to see this in more detail

Reproducibility 0 out of 1

As far as I could see, no codes were available (but I may be wrong and did not spend time checking this in detail)

---

### Decision · Program_Chairs · 2022-02-19

NA